# Sex-Dependent Changes in Risk-Taking Predisposition of Rats Following Space Radiation Exposure

**DOI:** 10.3390/life15030449

**Published:** 2025-03-12

**Authors:** Elliot Smits, Faith E. Reid, Ella N. Tamgue, Paola Alvarado Arriaga, Charles Nguyen, Richard A. Britten

**Affiliations:** 1EVMS School of Medicine, Macon and Joan Brock Virginia Health Sciences at Old Dominion University, Norfolk, VA 23507, USA; smitse@odu.edu (E.S.); redife@odu.edu (F.E.R.); 2EVMS Radiation Oncology, Eastern Virginia Medical School, Macon and Joan Brock Virginia Health Sciences at Old Dominion University, Norfolk, VA 23507, USAalvarapa@odu.edu (P.A.A.); charlesnewinski@gmail.com (C.N.); 3Center for Integrative Neuroscience and Inflammatory Diseases, Eastern Virginia Medical School, Macon and Joan Brock Virginia Health Sciences at Old Dominion University, Norfolk, VA 23507, USA

**Keywords:** space radiation, decision making, risk-taking propensity, sex dependency, processing speed

## Abstract

The Artemis missions will establish a sustainable human presence on the Moon, serving as a crucial steppingstone for future Mars exploration. Astronauts on these ambitious missions will have to successfully complete complex tasks, which will frequently involve rapid and effective decision making under unfamiliar or high-pressure conditions. Exposure to low doses of space radiation (SR) can impair key executive functions critical to decision making. This study examined the effects of exposure to 10 cGy of Galactic Cosmic Ray simulated radiation (GCRsim) on decision-making performance in male and female rats with a naturally low predisposition for risk-taking (RTP) prior to exposure. Rats were assessed at monthly intervals following SR exposure and the RTP performance contrasted with that observed during the prescreening process. Exposure to 10 cGy of GCRsim impaired decision making in both male and female rats, with sex-dependent outcomes. By 30 days after SR exposure, female rats became more risk-prone, making less profitable decisions, while male rats retained their decision-making strategies but took significantly longer to make selections. However, continued practice in the RTP tasks appeared to reduce/reverse these performance deficits. This study has expanded our understanding of the range of cognitive processes impacted by SR to include decision making.

## 1. Introduction

NASA’s decision to return to the Moon and eventually journey to Mars required a thorough assessment of the potential health risks that astronauts may face during long-duration missions. Space radiation (SR) is one flight hazard that could adversely impact astronaut health [1]; the predicted SR exposure levels for astronauts on missions to Mars may be as much as 1–1.2 Sv (~200 cGy) [1]. Ground-based rodent studies indicate that exposure to low SR doses (1–25 cGy) impairs performance in various cognitive processes (reviewed in [2,3,4,5,6]).

The Artemis missions will be particularly challenging due to the high cognitive demands placed on astronauts, who will need to make rapid, critical decisions in unfamiliar or high-pressure situations. Decision making is a complex process involving the rapid assessment of problems, weighing options, and selecting the most appropriate action after considering potential risks and benefits. It is concerning that exposure to <10 cGy of SR has been shown to significantly impair executive functions related to decision making, such as attentional set shifting (ATSET) [7,8,9,10,11,12] and task switching [13,14]. Executive functions also regulate impulse control and affect motivation, which, if impaired, can lead to maladaptive risk behaviors [15,16,17,18]. Therefore, SR exposure could result in both impaired decision making and aberrant behaviors, potentially exacerbating issues stemming from challenges like confinement and isolation.

Previous research has shown that female rats exposed to SR exhibit an increased risk-taking propensity (RTP) in a rodent version of the Balloon Analogue Risk Task (BART) [19]. In humans, performance in the BART task correlates with risk propensities in multiple situations [20,21,22,23]. Like the BART, the RTP task is a free-choice (decision making) test, where rats evaluate the risks and rewards of four available options and estimate the consequences of each [16,17]. SR exposure not only led to increased affective behavior, with rats choosing high-risk, well-rewarded options, but also slowed down the decision-making process.

While these rodent RTP data raise concerns about SR’s effects on decision making, the untrained rats used in the previous study [19] may not be an appropriate model for astronauts. NASA astronauts, by contrast, are highly trained individuals with superior decision-making skills and inherently low RTP. Furthermore, the rats in the prior study had no experience with the RTP task and had to learn its rules after being exposed to SR. Previous research suggests that SR exposure does not impact performance on tasks with which rats are already familiar [10], but impairs performance in situations where rats must learn new rules or use transitive inference. Therefore, the increased RTP observed following SR exposure in the cognitively naive rats [19] might reflect an inability to evaluate or apply the consequences of their choices effectively [24].

This study was designed to investigate the impact of SR exposure on RTP in male and female rats that were preselected for low RTP prior to exposure. RTP performance was measured before and at 30, 60, and 90 days after exposure to 10 cGy of Galactic Cosmic Ray simulator (GCRsim) radiation, which mimics the primary and secondary GCR fields that human organs will be exposed to on deep-space missions [25].

## 2. Materials and Methods

### 2.1. Regulatory Compliance

All procedures approved by the Institutional Animal Care and Use Committees of Eastern Virginia Medical School (EVMS) and Brookhaven National Laboratories (BNL, Upton, NY, USA) were compliant with the National Research Council’s “Guide for the Care and Use of Laboratory Animals (8th Edition)”.

### 2.2. Rat Demographics, Husbandry, Experimental Timeline, and Exercise Regimen

Wistar rats (Hla^®^(WI)CVF^®^; Hilltop Lab animals, Inc., Scottsdale, PA, USA) were used in this study. Rats were ~3 months old upon arrival at EVMS, with an average weight of 250 g. The timeline and age of the rats during various experimental procedures relative to the SR exposure are outlined in Table 1. At the time of SR exposure, the rats were 5–6 months old, which translates to a biologically equivalent human age of ~18 years old [26].

The rats were paired-housed in individually ventilated cages (Green Line Techniplast, Buguggiate, Italy), maintained on a reversed 12:12 light/dark cycle, and given ad libitum access to Teklad 2014 chow. Rats were implanted with ID-100 us RFID transponders (Electronic Devices, Santa Barbara, CA, USA) and maintained on an exercise regimen (30 min at 25 m/min, twice a week) for the entire duration of the study, except when the rats were housed at BNL.

### 2.3. RTP Performance Screening

The RTP test is an appetitive assay and is thus reliant upon the motivational status of the rat to find a food reward. To increase this motivation, rats are placed on food restriction (before and during the testing process) so that an individual rat’s weight is maintained at ~85% of its pre-food-restriction weight.

All RTP procedures were conducted in Bussey-Saksida rat touchscreen chambers ((Model 80604), Lafayette Instruments, Lafayette, IN, USA). Testing was conducted during the dark cycle, with the first rat being tested ~2 h into the 12 h dark cycle (Zeitgeber T + 2). The time at which testing was commenced was kept constant for an individual rat.

#### 2.3.1. Stimulus Response Training (STR)

After habituation to the touchscreen chambers, the rats were subjected to stimulus response (STR) training, which consisted of four stages (habituation, STR15, STR4, and STR1). In the STR15 stage, the rat gains a food reward by touching any of the 15 illuminated holes (within a 3 × 5 grid). In the STR4 stage, the rats can gain a food reward by selecting any of the holes within a block of four (2 × 2) lit holes, which are randomly located at one of eight locations. The position of the block is changed after any response (i.e., the correct selection of a lit hole or incorrect selection of an unlit hole).

During the STR1-timed stage, rats gain a food reward if they touch a single illuminated hole within 30 s. The location of the illuminated hole is changed after any response (correct or incorrect) to a random location within the grid. In the STR1-fast timed stage, a reward is given if the illuminated light is selected within 10 s of its appearance.

Training sessions occurred daily, with the rats given a maximum of 50 trials/session to reach criterion. Once rats reached criterion in a stage (Table 2), they proceeded to the next stage the following day. Rats that did not reach criterion within the permitted number of sessions (Table 2) were removed from the study.

#### 2.3.2. Risk-Taking Propensity (RTP) Task

The RTP task used in this study [19] differs from an established rodent gambling task [27] due to the introduction of an activation step (the rat must press a green activation light (situated in the middle hole of the 3 × 5 grid)) to activate the RTP task after each trial.

The RTP task utilizes four response lights, which each have a defined a win/loss probability, reward size, and loss penalty (Table 3). Rats have to select a hole within 10 s, and failure to do so ends the trial. The rats were given a maximum of five daily sessions (50 trials per session) to reach criterion (i.e., the selection of the “safe holes” (#6 or #9) in >66% of trials, with a minimum number of 30 trials, in two consecutive sessions).

### 2.4. Irradiation Procedure

Twenty-four male and 13 female rats satisfied our inclusion criterion and were shipped to BNL. Thirteen male and eight female rats were exposed to 10 cGy “simplified 5-ion” GCRSim at the NASA Space Radiation Laboratory (NSRL), at an overall dose rate of 0.5 cGy/min (~20 min exposure). The Galactic Cosmic Ray simulator (GCRsim) approximates the primary and secondary GCR fields that human organs within a deep-space vehicle will be exposed to [25]. The simplified 5-ion beam provides a uniform dose distribution within the bodies of both rats and mice and consists of the following ions (protons (1 GeV/n): 35%; ^28^Si (600 MeV/n: 1%; ^4^He (250 KeV/n): 18%; ^16^O (350 MeV/n): 6%; ^56^Fe (600 KeV/n): 1%; and protons (250 MeV/n): 39%). The radiation is delivered in the same sequence in which the ions are listed.

After the rats were transported back to EVMS, they were rehoused under the same conditions as described above. At 30, 60, and 90 days after irradiation, the rats were placed on a restricted diet for two days and reassessed in the RTP testing until they reached criterion or for a maximum of five consecutive days. Following the completion of RTP testing, the rats were returned to ad libitum rat chow until the next scheduled assessment.

### 2.5. Statistical Methods

Due to the frequent observation of gamma-distributed data following space radiation exposure, our data were statistically evaluated using the Mann–Whitney U test (two-sided) or Fisher’s exact test (two-sided). All statistical calculations were performed using the appropriate software program within Prism 10.2 (GraphPad Software, San Diego, CA, USA).

To provide more insight into the population distribution of the data, violin plots (which use kernel density estimation (KDE) to approximate a probability density function to visualize the distribution of quantitative data) were used in this study.

## 3. Results

Two rats had to be removed from the study: one female sham rat was euthanized after the 30-day assessment due to an intestinal contortion, and one male GCRsim-exposed rat was lost prior to the 90-day post-exposure assessment due to a degloving accident.

### 3.1. Pre-Exposure (Task Engagement and Learning Proficiency)

The prescreening process identifies rats that are both motivated to perform in the task and can identify the most profitable hole selection strategy. On average, the rats in this study reached criterion within three sessions (males: 2.87 ± 0.31; females: 2.79 ± 0.23). The selected rats were highly motivated to perform, completing >45 (out of a maximum of 64) trials across all the preselection sessions. The frequency with which male and female rats selected the best rewarded options was 82.9 ± 1.32% and 77.5 ± 1.85%, respectively. There were no significant differences in the learning and motivation statuses of the rats randomized to the sham or GCR-exposed cohorts in either sex (Table 4).

### 3.2. Pre-Exposure Processing Speed (Reaction Time)

The RTP used in this study requires that the rats actively restart the task after a trial, irrespective of whether the trial resulted in a loss (timeout) or win. Thus, the measured reaction times reflect the processing speed of the decision-making process. There were no apparent differences in the time that it took the rats to indicate a response in a trial between the rats assigned to the sham and GCRsim-exposed cohorts (Figure 1).

### 3.3. Post-Exposure Selection Choices

At 30 and 60 days post-exposure, there were no significant changes in the hole selection strategy of sham male rats from that used in the prescreening sessions (Figure 2A). All rats selected one of the low-risk/profitable choices in >65% of the trials. However, at 90 days post-exposure (during the third RTP assessment), two of the 11 sham rats selected the riskier options (holes 7 and 10) at a significantly (*p* < 0.003, Fisher’ exact) higher rate than they did previously.

Overall, the GCR-exposed rats did not exhibit any significant change in the hole selection strategy over the 90-day testing period. However, a single rat selected the higher-risk choices at a significantly (*p* < 0.001, Fisher’ exact) higher frequency (>60% of the trials) during both the 60- and 90-day assessments (Figure 2B).

Overall, the hole selection strategy of the sham female rats remained above the pre-exposure performance threshold of ≥66% of low-risk choices (Figure 3A). However, three of the eight GCRsim-exposed female rats had selection rates below this threshold (Figure 3B). At 30 and 60 days post-exposure, two rats (25%) selected the higher-risk options at an equal or higher rate compared to the profitable options, which was significantly (*p* < 0.01, Fisher’ exact) lower than the threshold rate. However, during the 90-day post-exposure (the third) assessment, the safe hole selection rate was close to the prescreening threshold level.

### 3.4. Post-Exposure Processing Speed

An analysis of the response time data from the first day of the 30-day post assessment provides a measure of the inherent impact of SR on the processing speed; subsequent assessments will be a net result of the processing speed loss, possibly offset by practice effects. GCR-exposed male rats had significantly longer (shams: 2.43 ± 0.28 s; GCR: 4.38 ± 0.74 s; *p* = 0.03, Mann–Whitney) response times than shams (Figure 4). Due to the requirement that the rats had to actively restart the test (after either a win or a loss), this increased response time was not a manifestation of the motivation of the rats to participate in the task.

A similar analysis of the response time/processing speed in the female rats did not reveal any significant differences in the processing speed in sham or GCR-exposed rats.

## 4. Discussion

While all space flight hazards pose a potential problem to the long-term health of astronauts, the deleterious effects of SR are the least characterized. Ground-based rodent studies suggest that SR exposure has a significant impact on several cognitive processes and behaviors. The present study has extended the range of processes impacted by SR to include decision making/RTP.

In both male and female rats, exposure to 10 cGy GCRsim resulted in significant impairments in decision making in the RTP task. Within 30 days of SR exposure, both male and female rats had altered performance in the RTP. In female rats, this was manifested as 25% of rats making less profitable decisions than they did prior to SR exposure. In contrast, SR-exposed male rats generally maintained the same low-risk selection rate but took significantly (~two-fold) longer to select than did the shams. There is an ever-growing number of studies that report different sex-dependent cognitive outcomes after SR exposure [7,8,13,14,28,29,30,31,32,33,34,35], and, presently, it would appear that the ultimate outcome of SR exposure in male and female rodents is highly dependent upon the cognitive domains being investigated. While the different SR-induced cognitive impairments observed in male and female rats may be related to SR’s effects on sex hormones, recent data suggest that the explanation for the apparent sex effects on executive functions may be far more complex. Imaging studies have identified sex differences in regional brain activity and distinct network activation during task performance [36,37]. It has also been suggested that sex differences in strategy and outcome assessment, which are critical aspects of learning, rather than innate sex differences in the underlying neurophysiology [38], determine performance in executive function tasks.

The increased RTP of the GCR-exposed female rats (Figure 3) is consistent with our previous studies with He-exposed rats [19]. However, in these earlier studies, the increased RTP could have been attributed to SR impacting rule learning, as the rats had no prior experience with the RTP assay. In contrast, the present study involved rats that were not only familiar with the RTP assay but had also identified the most effective hole selection strategy. Thus, the SR-induced increase in RTP seen in the current study (Figure 3) suggests that SR exposure impacts some aspects of the decision-making/evaluation process or possibly their long-term-working memory of the task. Although only 25–33% of the female rats showed impaired performance, this proportion approaches the threshold generally recognized as sufficient to compromise the combat effectiveness of a unit [39].

While the BART is generally considered to be a valid measure to assess RTP in humans, without additional independent measures of “real-life” risk taking in rats, it is dangerous to speculate about whether the observed SR changes in RTP in female rats reflect a change in risk taking. Specific experiments designed to establish the relationship with decision-making performance and “reckless” behavior are needed before such assumptions can be made. However, our data suggest that SR does impact some aspect of decision making in female rats. We have previously shown that GCRsim exposure impacts ATSET performance, a critical component of decision making. Moreover, this study suggests that SR adversely affects numerosity (and possibly abstraction) since performance in the RTP task requires that rats establish the probabilities and risks associated with the four presented options and estimate the consequences of each option. Rats have well-developed numerosity, abstraction, and metacognitive abilities [40,41]. However, specific experiments using the tasks employed in these studies will be required to establish the incidence and severity of SR-induced losses in these cognitive domains.

Whatever the reasons underlying these decision-making performance changes in irradiated female rats, they do seem reversible, since, by 90 days post-exposure (i.e., during the third post-exposure testing), the RTP performance decrements were no longer considered significant. Thus, the female rats appear to have either “passively” (e.g., regained their long-term working memory of the optimum reward strategy) or “actively” (relearned/reevaluated the profitability of the various section strategies) reversed the performance decrements.

In contrast, most male rats did not exhibit any alteration in the decision-making strategy after GCR exposure. However, during the first RTP assessment (at 30 days after exposure), there was a highly significant increase in the time that the rats took to make a selection (Figure 3). While most of the rats continued to make “profitable” selections during the 90-day (three tests) period, one rat abandoned the profitable selection approach at 60 and 90 days and selected the high-risk options in over 70% of the trials. However, this may not be related to SR exposure, since some sham rats adopted similar changes in selection strategy in the third post-exposure test, possibly due to boredom in the task.

The male data provide further evidence that SR exposure impacts processing speeds in complex tasks like ATSET [7,8,42] and task switching [14]. The underlying cause of the SR-induced loss of processing speed is presently unclear, but it is often observed in patients receiving cranial X-ray or proton therapy [43,44]. SR exposure results in task-switching costs [14], a phenomenon negatively correlated with processing speeds [45,46]. The processing speed can be a reflection of working memory [47,48,49,50], which is also impaired by SR exposure [3,4,5,51]. The processing speed is also impacted by neuronal myelination levels [52,53], which are significantly altered by SR exposure in mice [54].

## 5. Conclusions

In summary, our studies reaffirm that SR exposure affects cognitive performance in both male and female rats, although the nature of these impairments is highly context-specific. These findings are particularly concerning as they indicate that SR exposure can alter RTP predisposition in rats that previously demonstrated working knowledge of the task and a low predisposition for risk taking. Translating ground-based rodent studies into tangible risk estimates for astronauts remains an enormous challenge, but our data raise the possibility that astronauts may need to be closely monitored for alterations in decision-making/risk-taking behavior and/or loss of processing speed during cognitive loading. To mitigate these potential risks, it may be prudent for NASA to enhance its training programs to ensure that astronauts can effectively perform critical tasks and make sound decisions even if their executive functions are compromised by SR exposure. Training scenarios that simulate high-risk, high-pressure situations with limited decision-making time could help astronauts to prepare to respond more effectively under cognitive stress.

## Figures and Tables

**Figure 1 life-15-00449-f001:**
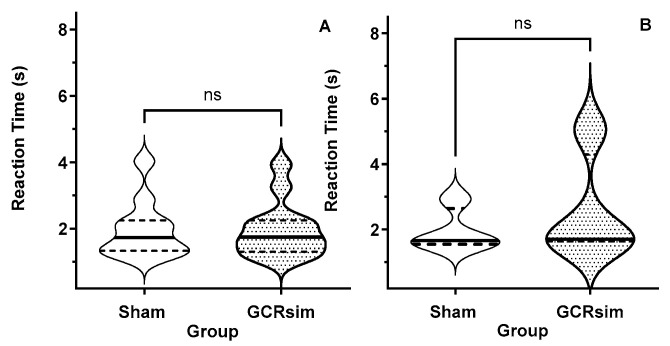
Violin plot of the reaction time (processing speed) during the prescreening RTP assessment in rats ((**A**) male; (**B**) female) that were later assigned as sham (open) or 10 cGy GCRsim-exposed (stripped) rats. The solid horizontal line represents the median, while the dashed lines represent the quartiles. ns denote “not significant”.

**Figure 2 life-15-00449-f002:**
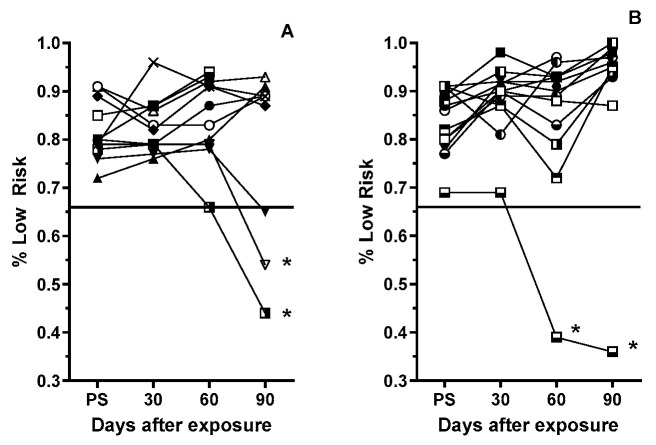
Frequency with which sham (**A**) and 10 cGy GCRsim (**B**) male rats chose the two optimally rewarded (“safe”) options in the prescreening and post-exposure RTP assessments. Symbols represent individual rats with connecting lines to aid visualization of individual rat performance. The solid horizontal line represents the prescreening threshold performance for inclusion in the study. * denotes a significant difference (*p* ≤ 0.01, Fisher’s exact) from the individualized prescreening performance.

**Figure 3 life-15-00449-f003:**
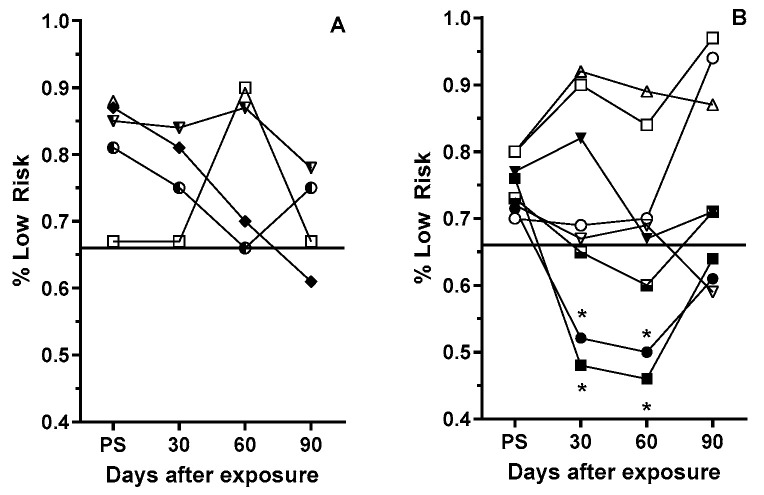
Frequency with which sham (**A**) and 10 cGy GCRsim (**B**) female rats chose the two optimally rewarded (“safe”) options in in the prescreening and post-exposure RTP assessments. Symbols represent individual rats with connecting lines to aid visualization of individual rat performance. The solid horizontal line represents the prescreening threshold performance for inclusion in the study. * denotes a significant difference (*p* ≤ 0.01, Fisher’s exact) from the individualized prescreening performance.

**Figure 4 life-15-00449-f004:**
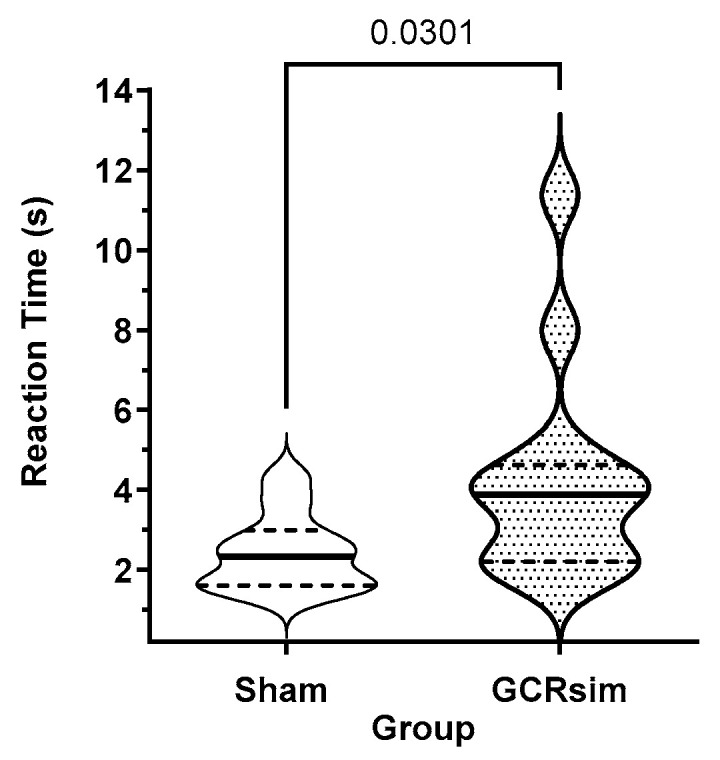
Violin plot of the reaction times for male sham (open) and 10 cGy GCRsim-exposed (stripped) rats in the first RTP task assessment performed at 30 days post-exposure. The solid horizontal line represents the median, while the dashed line represents the quartiles. Numbers above the comparison bar represent the statistical significance between the cohorts when analyzed using the Mann–Whitney test.

**Table 1 life-15-00449-t001:** Experimental timeline (relative to SR exposure).

Event	Time Relative to SR Exposure	Rat Age
Arrival at EVMS	~12 weeks pre	~3 months
Exercise	~11–3 weeks pre	
Ship to BNL	1 week pre	
SR exposure	-	~6 months
Ship to EVMS	1 week post	
Exercise	1–14 weeks post	
RTP testing	14–18 weeks post	~10 months

**Table 2 life-15-00449-t002:** Criterion during stimulus response (STR) training.

Stage ^1^	Response Window	# Correct Selections	Completion Rate	Permitted # Sessions	Days at Criterion
STR15	N/A	≥30	≥60%	8	1
STR4	N/A	≥30	≥60%	8	1
STR1-Timed	30 s	≥30	≥75%	8	2
STR1-Fast	10 s	≥30	≥75%	10	2

^1^ All sessions in all stages had a maximum duration of 30 min.

**Table 3 life-15-00449-t003:** Reward characteristics for the four options in the RTP Task.

Hole	Reward Size ^1^	Win Rate	Loss Time Penalty	Possible Reward ^2^
6	1	90%	5 s	216
7	4	40%	35 s	102
9	2	80%	10 s	320
10	3	50%	20 s	158

^1^ Number of sucrose pellets. ^2^ Number of sucrose pellets obtained during if always selected.

**Table 4 life-15-00449-t004:** RTP performance metrics.

	Male				Female			
	Sham		GCR		Sham		GCR	
TEST ^1^	STRC ^2^	Trials ^3^	STRC	Trials	STRC	Trials	STRC	Trials
Pre	3.09 ± 0.41	56.9 ± 2.9	2.69 ± 0.45	55.1 ± 4.4	2.2 ± 0.20	44.3 ± 2.43	3.12 ± 0.29	49.7 ± 1.51
Post + 30	2.63 ± 0.31	55.6 ± 3.0	2.69 ± 0.33	51.7 ± 2.9	3.00 ± 0.31	31.2 ± 2.59	2.51 ± 0.19	35.2 ± 2.9
Post + 60	2.45 ± 0.28	55.5 ± 2.5	2.38 ± 0.27	54.6 ± 2.6	3.1 ± 0.28	30.6 ± 2.5	2.75 ± 0.16	31.3 ± 2.25
Post + 90	3.11 ± 0.48	48.9 ± 3.9	2.27 ± 0.27	53.2 ± 2.6	3.5 ± 0.50	30.7 ± 3.5	3.03 ± 0.23	35.8 ± 2.3

^1^ Assessment performed relative to SR exposure. ^2^ Average number of sessions to reach criterion/completed. Values denote mean ± SEM. ^3^ Average number of trials completed per session. Values denote mean ± SEM.

## Data Availability

Dataset available on request from the authors.

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
