# Peer review of "Sex-Dependent Changes in Risk-Taking Predisposition of Rats Following Space Radiation Exposure"

_life, 2025, doi:10.3390/life15030449_

Round 1
Reviewer 1 Report
Comments and Suggestions for Authors
Dear Authors,
I have read your manuscript titled “Sex dependent changes in risk-taking predisposition of rats fol-2 lowing space radiation exposure” with great interest. The study addresses a highly relevant topic for the safety of long-duration space missions. However, there are aspects that should have been explored to strengthen the scientific robustness of the study.
1. The research provides interesting behavioural data but does not delve into the underlying biological mechanisms of radiation-induced damage. In particular, neuroinflammatory processes and the systemic effects of radiation, which may have repercussions on the brain, are not analysed and discussed. These aspects should be addressed both in the introduction and in the discussion, ideally supported by experimental data on inflammation markers and myelination.
2. The conclusion that the observed alterations are reversible is based solely on improvements in behavioural tests. However, since neurons do not regenerate significantly, the observed recovery might be attributed to the resolution of transient inflammation rather than a true functional restoration. It would be appropriate to support this hypothesis with biological data, such as the analysis of inflammatory cytokines and studies on myelination.
3. Differences between males and females are described, but there is no in-depth discussion of their biological basis. I suggest that the authors consider the possible influences of hormonal factors, neurochemical differences, and structural variations on the observed effects. Furthermore, the role of estrous cycles in females should be discussed, as hormonal fluctuations could influence risk-taking propensity and decision-making speed.
The study presents interesting data but would greatly benefit from a more comprehensive analysis of the neurobiological mechanisms underlying the observed behavioural changes.
Author Response
Reviewer 1
I have read your manuscript titled “Sex dependent changes in risk-taking predisposition of rats following space radiation exposure” with great interest. The study addresses a highly relevant topic for the safety of long-duration space missions.
We are pleased to read this.
However, there are aspects that should have been explored to strengthen the scientific robustness of the study.
- The research provides interesting behavioural data but does not delve into the underlying biological mechanisms of radiation-induced damage. In particular, neuroinflammatory processes and the systemic effects of radiation, which may have repercussions on the brain, are not analysed and discussed. These aspects should be addressed both in the introduction and in the discussion, ideally supported by experimental data on inflammation markers and myelination.
It is unclear why the reviewer is so adamant that neuroinflammation is the underlying cause of radiation-induced cognitive dysfunction in general, and for decision making in particular. It is true that neuroinflammation can be increased following radiation exposure, and a few studies that suggest that reducing the neuroinflammatory response “rescues” performance in very simple tasks, like NOR. In the case of NOR, a task that is almost exclusively dependent upon the hippocampus, there is some association between microglial activation in the hippocampus and NOR performance. However, complex executive functions (like decision making) require critical input from many brain regions, most of which are cortical. To date, there have not really been any studies that have assessed microglial activation in the various brain regions implicated in decision making. We could have indeed speculated on this one facet, but thought that the unbiased fMRI study (in humans) that identified the Corpus Callosum and IHWM tracts as exhibiting changes in activity that track with the phenotypic outcome we observe in the rats was more appropriate. We were unaware of these studies until we processed our data, and what brain regions we recovered did not include either of the implicated targets.
We hope that a reader of our study will now be in a position to do some targeted mechanistic studies. Access to the space radiation beam is extremely limited and follow-up studies need to be funded by NASA and have to be scheduled (often a year in advance).
- The conclusion that the observed alterations are reversible is based solely on improvements in behavioural tests. However, since neurons do not regenerate significantly, the observed recovery might be attributed to the resolution of transient inflammation rather than a true functional restoration. It would be appropriate to support this hypothesis with biological data, such as the analysis of inflammatory cytokines and studies on myelination.
We would agree that neurons, especially those in the multiple cortical regions that are involved in executive functions, have limited regenerative ability. Indeed, there is very little direct evidence that neuronal cell loss underlies radiation induced cognitive impairment. The significant loss of performance at doses of 10 cGy or less (and the almost complete lack of an LET-dependency) suggest that neuronal dysfunction rather than cell killing is more likely the cause of the cognitive impairment. We have published a commentary (Britten & Limoli, 2023 (in Life)) on this subject. The “restoration” of performance with repeated practice (and/or time since exposure) could be classified as reversal of the deficit. We carefully explained that this could be an active (or a passive) relearning of the rules of task. Practice effects are well recognized in many task behaviors and occur in normal individuals as well as those suffering from non-radiation induced brain trauma. Once again we have published a commentary of cognitive rehabilitation (Britten & Blackwell, 2022).
Given the generality of these phenomena, these seem far more likely to explain the “reversibility” of the cognitive defects, than the resolution of neuroinflammation. While we could speculate that this may be one explanation for our findings, for completeness we would also have to include changes to the blood-brain barrier, mitochondrial dysfunctions, change in neurotransmission (pre and post synaptic), changes in neurotransmission hierarchy. Our group has indeed conducted such studies as suggested, but given the multiple brain regions involved in decision-making, and our ignorance of the clinical studies implicating the corpus callosum, we did not conduct such studies in brain regions likely to be involved.
- Differences between males and females are described, but there is no in-depth discussion of their biological basis. I suggest that the authors consider the possible influences of hormonal factors, neurochemical differences, and structural variations on the observed effects. Furthermore, the role of estrous cycles in females should be discussed, as hormonal fluctuations could influence risk-taking propensity and decision-making speed.
We intentionally did not dwell on this facet of our study, but instead chose to simply outline our observations. We obviously did consider the factors outlined by the reviewer, and the many studies that clearly demonstrates intrinsic sex-dependent changes in the inherent response of the CNS during executive function task performance. There are also several studies suggesting that the economic cost tolerance of male and female rats differs, i.e., they may have intrinsically different motivations to gain a food reward. Loss of reproductive hormones can indeed impact many cognitive functions. Considering all the available data, and the overall lack of studies establishing the impact of SR on the cognitive performance (especially executive functions) in female rodents (and on ovarian function), we decided to not speculate on the underlying difference observed in the male and female rats. To simply attribute these effects to circulating gonadal hormones would be naïve and irresponsible.
However, we were negligent in not outlining the data that suggests that executive function performance may be regulated by a different balance of brain regions in male and females. We thank the reviewers for noting this oversight.
We have added the following test in the Discussion.
“While the different SR-induced cognitive impairments observed in male and female rats may be related to SR-effects on sex hormones, recent data suggest that the explanation for apparent sex effects on executive functions may be far more complex. Imaging studies have identified sex differences in regional brain activity and distinct network activation during task performance [36][37]. It has also been suggested that sex differences in strategy and outcome assessment, critical aspects of learning, rather than innate sex differences in the underlying neurophysiology [38] determine performance in executive function tasks.”
The study presents interesting data but would greatly benefit from a more comprehensive analysis of the neurobiological mechanisms underlying the observed behavioural changes.
We would agree in part with the reviewer, but such studies were not within our mandate for the grant, and given the strongly implicated role of the Corpus callosum, we hope that in future, such investigations can now be targeted to appropriate areas.
Reviewer 2 Report
Comments and Suggestions for Authors
The authors present a manuscript entitled “Sex dependent changes in risk-taking predisposition of rats following space radiation exposure” in which they show possible correlations between decision-making ability and exposure to cosmic radiation-like conditions, focusing specifically on sex-dependent differences.
The authors should note the following:
The authors show significant errors when presenting the uncertainty and significant figures of the quantities (e.g. in table 4). It is recommended to follow the guidelines described in international standards, in particular JCGM 100:2008 (Evaluation of Measurement Data - Guide to the Expression of Uncertainty in Measurement (GUM 1995 with minor corrections), Joint Committee for Guides in Metrology, 2008).
The authors show the results in violin plots (Figure 1 and Figure 4), but the statistical descriptions of these results are not sufficiently developed; they do not describe why the asymmetry shown in the images occurs, nor the meaning of the bias shown by the deviation of the response time towards higher values. A solid horizontal line appears which must be the median, while the dashed lines must be the quartiles (in the parts related to GCRsim these lines are practically not observed), but the authors make no reference to this. Figures 2 and 3, on the other hand, are very difficult to understand, they are not sufficiently explained.
Between lines 41 and 45 of the introduction, the authors provide a comprehensive overview of previous results, supported by numerous references. However, it is not clear what novel contributions this study offers. Clarifying the unique aspects of this work would enhance its scientific impact.
In section “2.4. Irradiation procedure”, the authors briefly indicate the laboratory where the irradiation was carried out, but in addition to this, they could develop a little more the irradiation methodology implemented, indicating for example dose rate, energy, etc.
Comments on the Quality of English Language
I don't feel qualified enough to evaluate the quality of English.
Author Response
Reviewer 2
The authors present a manuscript entitled “Sex dependent changes in risk-taking predisposition of rats following space radiation exposure” in which they show possible correlations between decision-making ability and exposure to cosmic radiation-like conditions, focusing specifically on sex-dependent differences.
The authors should note the following:
The authors show significant errors when presenting the uncertainty and significant figures of the quantities (e.g. in table 4). It is recommended to follow the guidelines described in international standards, in particular JCGM 100:2008 (Evaluation of Measurement Data - Guide to the Expression of Uncertainty in Measurement (GUM 1995 with minor corrections), Joint Committee for Guides in Metrology, 2008).
It is unclear what the reviewer is asking for in this comment. Presenting data as a mean with the standard error of the mean, as a metric of variability (variance) within the sample population, is a fairly standard practice (for biological studies). We use 2-sided variants of the statistical algorithms, and we favor using Mann-Whitney as it is non-parametric and thus not dependent upon normally distributed data.
The authors show the results in violin plots (Figure 1 and Figure 4), but the statistical descriptions of these results are not sufficiently developed; they do not describe why the asymmetry shown in the images occurs, nor the meaning of the bias shown by the deviation of the response time towards higher values.
A central facet of our study (and many of our preceding studies) is that a sub-population of individuals exhibit marked SR loss of performance, while others show no obvious loss of performance, i.e., inter-individual variability. The use of violin plots is increasingly being used, as it provides considerably more information about the distribution of the data points. Most “perturbed” data sets are gamma distributed, and thus gaussian based statistics (including mean +/- SD or SEM) are not optimal, either for analysis of depiction of the data. Violin plots typically use Kernel density estimation (KDE) approaches to calculate the probability density of the data. Since the data is often skewed, the quartiles are often asymmetrical. We assumed that the readers of a reputable journal like Life, would be familiar with such analysis and thought that it would be insulting to their intelligence to cover such basic points. However, the reviewer’s comments suggest that we should have at least included a statement on the benefits of using KDE analysis, and portraying the data as a violin plot since this provides a nice representation of the actual distribution of the data within the population. We have added a few sentences in the statistical Methods to cover this point.
A solid horizontal line appears which must be the median, while the dashed lines must be the quartiles (in the parts related to GCRsim these lines are practically not observed), but the authors make no reference to this. Figures 2 and 3, on the other hand, are very difficult to understand, they are not sufficiently explained.
We thank the reviewer for this comment. The legends for both figures 1 and 2 clearly state that the solid horizontal line depicts the median, while the dashed lines represent the quartiles. The shading of the GCRsim symbol did obscure the “quartile” lines and we have changed the shading pattern so that these are more clearly visible. In Figure 1, the lower quartile is indeed very close to the median.
Between lines 41 and 45 of the introduction, the authors provide a comprehensive overview of previous results, supported by numerous references. However, it is not clear what novel contributions this study offers. Clarifying the unique aspects of this work would enhance its scientific impact.
This is a somewhat puzzling comment. The cited studies pertain to specific processes that are involved (actually stated that they are “related to”) in the decision making process, but the tasks employed in those studies (ATSET and task switching) do not look at decision-making in its entirety. That process also involves psychological processes (like motivation, affect and impulsivity). The BART analog used in the present study, investigates the entire decision making process, not just ATSET and task switching processes. We believe that the following three paragraphs outline the novelty of the present study.
In section “2.4. Irradiation procedure”, the authors briefly indicate the laboratory where the irradiation was carried out, but in addition to this, they could develop a little more the irradiation methodology implemented, indicating for example dose rate, energy, etc.
Both Reviewer 2 and Reviewer 3 have raised this important point. To reduce the “duplicated” content from other papers, a more detailed description of the GCRsim beam had to be removed. We agree that it is important for the reader to know that this beam was carefully designed to correspond to the spectrum (energy/LET) of particles that the internal organs of astronauts will be exposed to. In light of the reviewers’ concerns we have happily added a better conceptual description of this unique radiation sources, and its specific constitutive ions (and energies).
Reviewer 3 Report
Comments and Suggestions for Authors
I read with great interest this relatively short article, which investigates certain effects of low-dose ionizing radiation exposure on decision-making performance in male and female rats. A challenging study, experimentally speaking. The results appear convincing, although, in my opinion and contrary to the authors' view, difficult to directly translate to humans. I recommend accepting the article.
The only clarification needed is the type and energy of the ionizing radiation used. What exactly is this "Galactic Cosmic Ray simulator radiation"? Is it a mix of different radiations? If so, which ones, their energies, and their LETs? Reference 25 is insufficient in this context.
Additionally, reference 24 appears incomplete.
Author Response
Reviewer 3
I read with great interest this relatively short article, which investigates certain effects of low-dose ionizing radiation exposure on decision-making performance in male and female rats. A challenging study, experimentally speaking. The results appear convincing, although, in my opinion and contrary to the authors' view, difficult to directly translate to humans. I recommend accepting the article.
Until astronauts are exposed to this type of radiation in space, and their decision-making performance assessed, it is hard to know if our data are translatable to humans. The BART test, which the rodent RTP task is an analog of, is used in humans. There are clearly many reasons why rodent studies may not translate to humans, the purpose of NASA’s ground based research program is not to generate risk estimates per se, but to identify “areas of concern” that may need to be monitored in astronauts. Due to “duplication” constraints many of the phrases that we would normally use to address the rodent/human transition has to be removed. We are pleased that you would still recommend accepting the manuscript.
The only clarification needed is the type and energy of the ionizing radiation used. What exactly is this "Galactic Cosmic Ray simulator radiation"? Is it a mix of different radiations? If so, which ones, their energies, and their LETs? Reference 25 is insufficient in this context.
See response to the same point raised by Reviewer 2.
Additionally, reference 24 appears incomplete.
We thank the reviewer for spotting this. Unfortunately, the citation manager program did not transfer all the details, when we applied the recommended format for this journal. We have now corrected this